# A Novel Agricultural Machinery Intelligent Design System Based on Integrating Image Processing and Knowledge Reasoning

Cheng'en Li [1], Yunchao Tang [2,*], Xiangjun Zou [1,3], Po Zhang [1], Junqiang Lin [1], Guoping Lian [4] and Yaoqiang Pan [1]

1. College of Engineering, South China Agricultural University, Guangzhou 510642, China
2. College of Urban and Rural Construction, Zhongkai University of Agriculture and Engineering, Guangzhou 510006, China
3. Foshan-Zhongke Innovation Research Institute of Intelligent Agriculture and Robotics, Foshan 528000, China
4. Department of Chemical Engineering, University of Surrey, Guildford GU2 7XH, UK
* Correspondence: ryan.twain@gmail.com; Tel.: +86-134-3395-5703

**Abstract:** Agricultural machinery intelligence is the inevitable direction of agricultural machinery design, and the systems in these designs are important tools. In this paper, to address the problem of low processing power of traditional agricultural machinery design systems in analyzing data, such as fit, tolerance, interchangeability, and the assembly process, as well as to overcome the disadvantages of the high cost of intelligent design modules, lack of data compatibility, and inconsistency between modules, a novel agricultural machinery intelligent design system integrating image processing and knowledge reasoning is constructed. An image-processing algorithm and trigger are used to detect the feature parameters of key parts of agricultural machinery and build a virtual prototype. At the same time, a special knowledge base of agricultural machinery is constructed to analyze the test data of the virtual prototype. The results of practical application and software evaluation of third-party institutions show that the system improves the efficiency of intelligent design in key parts of agricultural machinery by approximately 20%, reduces the operation error rate of personnel by approximately 40% and the consumption of computer resources by approximately 30%, and greatly reduces the purchase cost of intelligent design systems to provide a reference for intelligent design to guide actual production.

**Keywords:** agriculture machinery; knowledge base; deep learning; virtual prototype

## 1. Introduction

With the continuous improvement of agricultural machinery intelligent design databases, most parts can be redesigned and put into production by means of reuse design. It is the core of intelligent design and advanced manufacturing technology of key parts of agricultural machinery to build a virtual prototype with advanced technology and complete functions for production simulation and data analysis [1,2]. However, the current cost is too high for small and medium-sized agricultural machinery equipment enterprises to purchase large-scale intelligent design systems. In designing key parts of agricultural machinery that consist of numerous parts with high similarity, the calculation efficiency and the utilization rate of software function are low, and the coherence and flexibility are weak. Therefore, the construction of a virtual prototype and its unique knowledge base of key parts of agricultural machinery with strong functionality, high data abundance, low cost, and strong practicability is of great significance for the intelligent design of agricultural machinery to guide production [3–5].

In this work we analyze the existing intelligent design platform and research results. In terms of system functions, SolidWorks has released its commercial plug-in for collision detection. Users can select features and define the relative motion relationship between parts

to obtain interference data. However, there is a lack of global data automatic detection and a real-time feedback function of multiple parts. Rameau designed the calculation method of fit clearance under various assembly conditions [6]; however, the computational efficiency of this method was low, and the algorithm process was cumbersome for the agricultural machinery assembly data with strong regularity. Schleich designed a multifunctional virtual prototype for product design data analysis [4,7,8]. Assad constructed a simulation system to estimate production data [9–11], but they both lack the data compatibility of simulation analysis function and multiplatform interaction. In terms of data management, Ahmed built a cross-system product data management platform [11]; Wu built an algorithm for reuse data analysis and call [12,13]; Zhang designed the reconstruction algorithm of the feature surface to calculate the part feature data [14,15]; and Bobenrieth used 2D sketches for reuse design and model retrieval [16,17]. However, there are still problems of poor real-time and simulation. In terms of interactivity, NX, Creo, and other mainstream commercial CAD software import parts and manually plan assembly through users during virtual assembly; Wang, used algorithms to construct assembly constraints to control part assembly [18–21]; and Alejandro combined virtual reality technology to develop a more interactive and practical design system [4,22–24].

In summary, the above research still contains the problems of low system function coherence, low simulation degree, poor interaction, high cost, low calculation efficiency, lack of dynamic update of data, and real-time analysis ability [25]. Based on the existing research, this paper further combines the features of key parts of agricultural machinery and advanced technologies, such as image processing and deep learning, to detect the feature parameters of parts and construct its virtual prototype and verifies the data accuracy of the virtual prototype by the method of the curvature radius of contour points. A special knowledge base containing enterprise production process data is constructed, a variety of design tests and data analyses are conducted using a virtual prototype, and the reliability of the knowledge base data is verified by comparing the theoretical value with the actual value. The virtual prototype and knowledge base are integrated in the virtual reality engine, and an intelligent design system for key parts of agricultural machinery is developed. Finally, a practical production case is used to indicate the practicality of the system. The main contributions and innovations of the article are as follows:

(1) The system uses the image-processing algorithm, with higher measurement accuracy and trigger, to measure the characteristic parameters of the 3D model of parts and reconstructs the virtual prototype with the simulation appearance, collision, and interference detection function and simulation of the physical characteristics of the 3D model according to the parametric design process of mechanical parts.

(2) The system constructs a knowledge base that can be updated dynamically, infers and updates the data of assembly sequence, assembly benchmark, fit tolerance zone and product interchangeability of the virtual prototype in real-time, and carries out production tests and analyses of the test data. At the same time, the system can freely match the knowledge reasoning program of product tests according to the design requirements, and carry out data analysis flexibly, coherently, and automatically.

(3) According to the assembly logic of key parts of agricultural machinery, the system simplifies the assembly behaviour, uses VR equipment to complete the virtual assembly with stronger interactions and higher immersion, and uses the knowledge base to feed back the analysis results of assembly interference data and user assembly operation specifications in real-time. This helps to improve the efficiency of assembly training of small and medium-sized enterprises.

(4) Based on the virtual reality engine, this paper constructs an intelligent design system for key parts of agricultural machinery with more powerful real-time and calculated functions. Finally, the third-party software evaluation and practical production application test results show that the system is more in line with the actual production situation of small and multi-batch, and changeable production requirements of small

and medium-sized enterprises. It can effectively improve the design and production efficiency and reduce the production cost.

The structure of the article is as follows: the second chapter is the process of virtual prototype construction and accuracy verification; the third chapter is the construction of a knowledge base and the verification process of data reliability; the fourth chapter is the overall application process of the system and the analysis of a third-party software test and evaluation data; and the last chapter summarizes the advantages and disadvantages of the system.

## 2. Constructing Virtual Prototype

The traditional 3D model prototype obtains test data through various CAD test modules. Its essence is based on the numerical calculation and features data interaction of a single part model prototype, which lacks global real-time data feedback and an independent data calculation function. The virtual prototype constructed in this study uses the collision trigger mechanism of the mesh solid model to automatically feedback interference data and assembly data. The data storage and feedback of each part can be completed independently and synchronously through the data script, which has the advantages of real-time data calculation, independent feedback, and global synchronous analysis.

The object structure of the virtual prototype is a multilevel structure, including a mesh solid model layer for physical simulation and data acquisition, a data layer for accessing part of the CAD data, an assembly datum layer for assembly behaviour definition, and a 3D model layer for appearance rendering. The mesh solid model and data layers access and calculate data through scripts, which can be called by the main program of the system. The call logic of each part is shown in Section 3. The schematic diagram of its structure level and screenshot of the system are as shown in Figure 1.

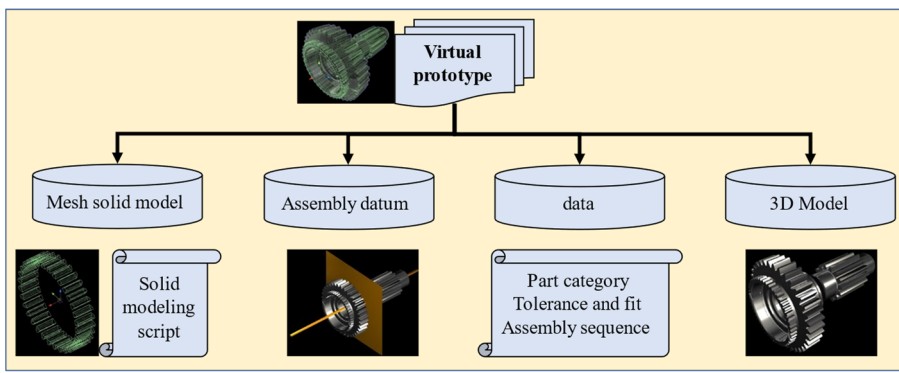

**Figure 1.** Virtual prototype.

### 2.1. Construction of the Mesh Solid Model Layer

To train the deep-learning network that recognizes the part category and the feature category contained in the part, we need to use the sample image data set of the model. The system needs to solve the problems of file format conversion, model size normalization, and model centroid coordinate system reconstruction of a large number of models, so that the image sample size of the model and the model attitude are both consistent. The mesh solid model layer refers to the mesh solid model composed of a collision bounding box and corresponding physical simulation components that fit the contour of the part. It has the functions of multibody continuous collision detection and triggering, simulation of physical properties, and so on. It is the basis for the system to obtain all test and analysis data. The built-in mesh bounding box attribute of the virtual reality engine only has the trigger bounding box fitting with the outer contour of the 3D model for the nonconvex 3D model but does not have the function of collision detection and the simulation of physical attributes of a rigid body. However, according to the parametric design process of mechanical parts, we can make the mesh bounding box have the attributes of a multibody

continuous collision trigger and simulation physics by slightly dividing it into a set of mesh bodies regularly spliced by a limited number of convex sub-meshes [2,25].

The architecture of this section is as follows: in Section 2.1.1, the system pre-processes the 3D model of key parts of agricultural machinery and constructs the image dataset for deep-learning network training [26]; in Section 2.1.2, based on the 3D model of the part, the system uses the image-processing algorithm to detect the key feature parameters of the part from the orthogonal view image; in Section 2.1.3, the system uses the differential method to construct the general mesh solid model of parts.

### 2.1.1. Training of the Feature Detection Network

First, we collect approximately 10,180 CAD three-dimensional models of key parts of agricultural machinery, convert their file formats in batches using the macro command of SolidWorks and 3DMAX, and import them into the virtual reality engine. Second, combined with the structural characteristics of the rotating body of the key parts of agricultural machinery, the system constructs the cube bounding box. According to the side length relationship of the bounding box, the posture of the parts is adjusted until the rotation axis is parallel to the x-axis of the world coordinate system and scaled to the appropriate size (see Figure 2). Through experiments, the best resolution of image samples in the dataset is $960 \times 960$, and the best size of parts in the sampling environment is 4. According to this conclusion, the system obtains the value of the rotation axis side length (L) and radial side length (R) of the cube bounding box of the three-dimensional model of the part and scales them into R = 4, L = L $\times$ 4/R [27].

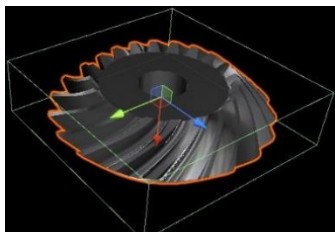

**Figure 2.** Cube bounding box and its datum axis.

At the same time, the shaft and general parts are divided according to the proportion of radial and axial dimensions of the parts.

Then, the system samples the parts one by one in the three orthogonal view directions of the top view, front view, and side view to obtain the image features of the rotating surface of the parts in the front view and the cross-section of the parts in the side view. It should be emphasized that the centre of the sample image corresponds to the projection point of the part centroid, which is the reference point for measuring the feature parameters and is used to describe the relative position relationship of each feature on the part [28]. Finally, the system constructs the image dataset for the training of each feature detection network [27].

### 2.1.2. Detection of Feature Parameters

The system uses the sample image data set constructed in Section 2.1.1 and trains the deep-learning network for classifying part categories and part feature categories; at the same time, OpenCV image-processing algorithm is used to detect the feature parameters of parts, and the goal of extracting feature parameters from part sample images is completed. The essence of the virtual sample mechanism construction of parts is the data reconstruction of key features of parts. The assembly features of key parts of agricultural machinery are clear and classified. The system uses an image-processing algorithm to accurately measure the feature parameters, makes full use of the feature classification function of the deep-learning network with high precision, and avoids the accurate data measurement function of deep-learning instability. Referring to the data expression of engineering drawings, the

feature data are measured from two-dimensional images, which improves the detection efficiency and accuracy.

Use of an Image Algorithm to Detect Feature Parameters

The key feature parameters of parts include the design parameters of mechanical parts such as tooth tip circle radius, number of teeth, tooth width, arc outer diameter, and arc inner diameter, for which the system sets corresponding detection means. Tables 1 and 2 summarize the feature combination of all key parts of agricultural machinery and the key parameters of each type of feature.

**Table 1.** Combination relationship between parts and various features.

| Part Type | Feature Combination Relationship |
| --- | --- |
| Cylindrical spur gear | Straight gear feature, Ring feature of inner hole, Internal spline feature |
| Cylindrical helical gear | Helical gear feature, Ring feature of inner hole, Internal spline feature |
| Spiral bevel gear | Spiral bevel gear feature, Ring feature of inner hole, Internal spline feature |
| Shift fork | Ring of fork ring, Ring feature of shaft hole |
| shaft | Ring feature, Straight gear feature, Spline feature, Helical gear feature |
| Meshing sleeve | Ring feature, Internal spline feature |
| axle sleeve | Ring feature, Internal spline feature |
| Bearing | Ring feature |

**Table 2.** The combination relationship between part features and feature parameters.

| Feature Type | Feature Parameter |
| --- | --- |
| Straight gear feature | Radius of addendum circle, Number of teeth, Tooth width |
| Helical gear feature | Radius of addendum circle, Number of teeth, Tooth width, Rotation direction, Helix angle |
| Spiral bevel gear feature | Radius of addendum circle, Number of teeth, Tooth width, Rotation direction, Helix angle, Large end modulus, Small end modulus, Cone angle |
| Ring of fork | Sector angle, Inner diameter, Outer diameter, Thickness |
| Spline feature | Radius of addendum circle, Number of teeth, Tooth width |
| Ring feature | Inner diameter, Outer diameter, Thickness |

The feature contour edge of the top view of the part can be summarized as a multi-segment square wave function. The edge of the contour of the part in the image sample is equivalent to the square wave function. Extracting the key feature parameters of parts can be equivalent to measuring and screening the discontinuities of function and slope. Based on the classification and positioning of the low-precision data of the feature detection network, we compare the positioning data of the image-processing algorithm with the data of the former to accurately locate and classify each feature and extract its parameters. For example, the centroid coordinate of a straight gear feature in the detection result of the feature recognition network is (500, 1440), and the system assigns its feature type result to the feature segment closest to (500, 1440) in the feature extracted by the image-processing algorithm. In addition, the unassigned feature segments are arc feature segments by default. This step avoids the positioning error of the feature recognition network (for example, all pixels of the feature are not completely selected in the right a priori box in Figure 3) and makes full use of its target classification function to judge the type of feature segment. Among them, the process of the image-processing algorithm is as follows:

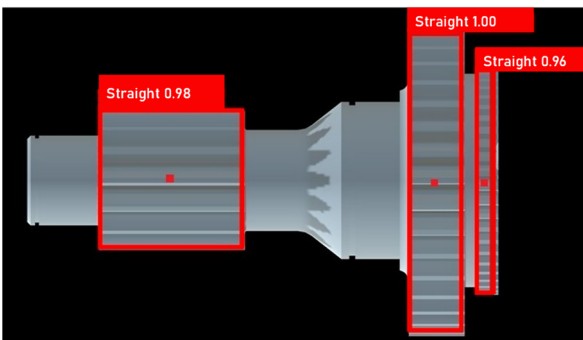

**Figure 3.** Feature recognition network classification feature.

It can be seen from the processing results that the feature contour edge of the top view of the part can be summarized into a multisegmented square wave function. The processing of the contour edge is actually the detection and screening of the discontinuity point of the square wave function and the band with zero or positive infinity slope.

$$\vec{P}_i = (x_i, y_i), \left( \vec{P}_i \in canny, y_i < \frac{Heigth}{2} \right) \tag{1}$$

$$|y_{i+1} - y_i| = \triangle y_i \tag{2}$$

$$Unit = \frac{Size \times 1000}{2880} \tag{3}$$

$$\left\{ \triangle y_i \leq K_Y \quad \left\{ \begin{array}{c} y_{i+1} = min\{y_i, y_{i+1}\} \\ R_j = \left( \frac{Height}{2} - y_{i+1} \right) \times Unit \\ S_j = S_j \cup \left\{ \vec{P}_{i+1} \right\} \end{array} \right\} \right. \tag{4}$$

$$\left. \triangle y_i > K_Y \quad \left\{ \begin{array}{c} J = J \cup \left\{ \vec{P}_{i+1} \right\} \\ R = R \cup \{R_{i+1}\} \\ j = j + 1 \end{array} \right\} \right\}$$

$$L_j = |J_{j+1} - J_j| \times Unit \tag{5}$$

$$\vec{C}_j = \frac{J_{j+1} - J_j}{2} \times Unit - \vec{O} \tag{6}$$

$$L_i \leq K_L, \left\{ \begin{array}{c} L = L - L_i \\ R = R - R_i \\ C = C - \vec{C}_i \end{array} \right\} \tag{7}$$

First, the system uses the Canny algorithm to extract all contour edge points (*canny*) in the upper half of the image with height (*Height*) and width of 2880 px (Formula (1)). At the same time, the conversion scale operator *Unit* between pixel units and millimeters is solved by the square field size (*Size*) of the virtual camera (Formula (3)).

Second, based on whether the pixel coordinate height difference $\triangle y_i$ of the adjacent point $\vec{P}_i, \vec{P}_{i+1}$ exceeds the threshold $K_Y$, the algorithm selects points P1 and P3 in Figure 4 as the starting point of the next feature contour and the end point of the current feature segment, respectively, and stores them in the set $J, S_j$, where the number $j$ of points set $J$ represents the number of feature segments; Set $S_j$ stores all contour edge points of each feature segment.

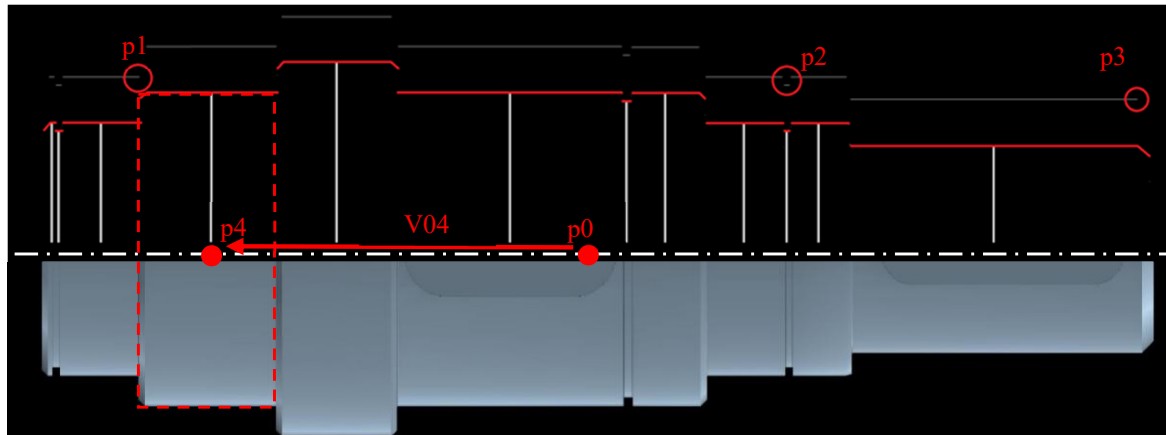

**Figure 4.** Accurate measurement of characteristic parameters by the image-processing algorithm.

At the same time, whenever the algorithm records a new segment of feature points, the minimum height of all points in $S_j$ is used as the contour radius $R_j$ of the current feature segment (see the white vertical line segment in Figure 4), the horizontal distance between the starting point and the last point is taken as the length $L_j$ of the feature segment, and the midpoint of the starting point and the last point is selected to solve the relative position relationship between it and the midpoint of the image $\vec{O}$ as the centroid $\vec{C_j}$ of the feature segment (see point P0 (1440, 1440) and vector V04 in Figure 4).

Finally, the algorithm filters the length $L_j$ of each feature segment according to the threshold $K_L$ and obtains the centroid relative coordinate set $C$, length set $L$, and radius set $R$ of all key feature segments.

Use of a Trigger to Detect Feature Parameters

The feature recognition network has poor accuracy in identifying a large number of features. The system uses the detection ball with a collision trigger function to detect the number of teeth, helix angle, rotation direction, shaft hole, internal spline, and other parameters of the parts. Among them, the radius of the unit sphere and the thickness of the detection disc are very small, and the trigger information between them and the part trigger can be returned. The schematic diagram of the test ball is as Figure 5.

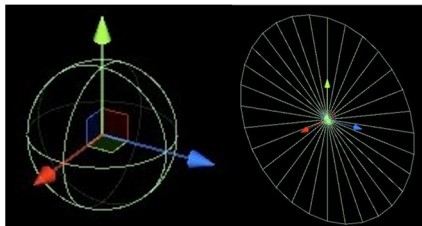

**Figure 5.** Detection ball and detection discs.

$$R_j = 0.98 \times H_j \tag{8}$$

$$\theta_r = 2arcsin\left(\frac{r}{R_j}\right) \tag{9}$$

$$\vec{P_{i+1}} = M_x(\theta_r)\vec{P_i} \tag{10}$$

(1) Process of detecting the number of teeth: the system controls the detection ball to move according to the track in Figure 6a, records the trigger times, and solves the teeth.

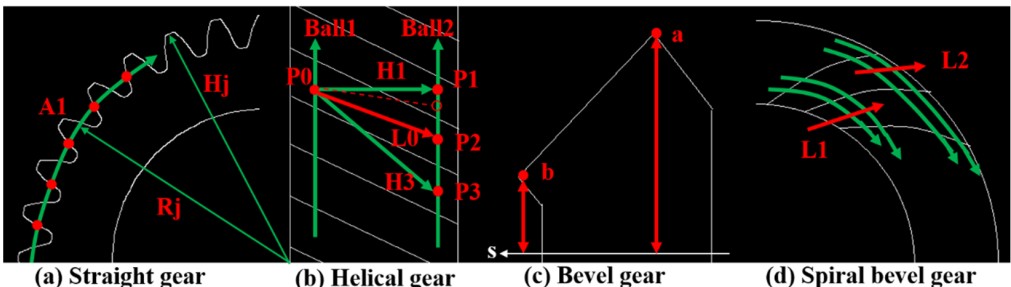

**Figure 6.** Gear feature parameter detection.

where $R_j$ is the radius of the track of the detection ball. $\theta_r$ is the rotation angle of the detection ball per frame. $H_j$ is the radius of the addendum circle. $r$ is the radius of the detection ball; $\vec{P_i}$, $\vec{P_{i+1}}$ is the position of the detection ball before and after rotating for one frame. $M_x(\theta_r)$ is the rotation matrix about the x-axis.

$$tan\beta = R \times \frac{tan\beta_k}{R_k} \tag{11}$$

(2) Process of detecting the helical angle and direction of the helical gear: As shown in Figure 6b, we take the midpoint P0, P1, P2, and P3 of the adjacent trigger points in the detection ball tracks Ball1 and Ball2 on both sides and combine them into paths H1, L0, and H2. Among them, P1, P2, and P3 are obtained from the adjacent points on the left and right sides of the projection point of P0 on the ball2 track. The helix corresponding to the path without a trigger can be solved using Formula (11). $\beta$ is the slope corresponding to L0, and $\beta_k$ is the helix angle obtained by the conversion relationship between the track radius $R$ and the dividing circle radius $R_k$.

(3) Process of detecting spiral angle and direction of spiral bevel gear: The target detection algorithm and image-processing algorithm locate and extract the maximum and minimum radius point B and point s of the conical tooth feature contour on the image and solve the radius of the tooth top circle at the large end and small end, sub cone angle, and tooth width. According to the algorithm in (2), we solve the spiral angle (see L1 and L2 in Figure 6c,d) and direction of the large end and small end of the gear.

(4) Process of detecting the shift fork: The image-processing algorithm obtains the positions of the shift fork ring and the shift fork shaft hole from the deep-learning target detection network and obtains the fork ring attachment point P1. P2 is triggered by the movement of the detection ball along the axis of the shift fork hole, and P3 is triggered by the movement along F2 again and solves the radius and relative position of the fork ring (see Figure 7).

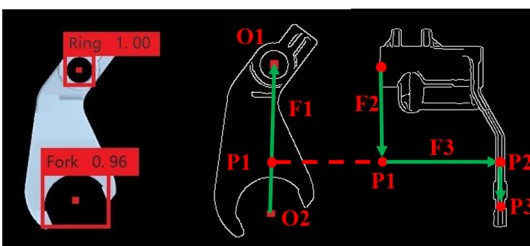

**Figure 7.** The mesh solid model and the detection sequence of the shifting fork.

(5) Process of detecting inner contour feature segments: The system controls the two detection discs, D1 and D2, to move along the rotation axis direction (the direction of the white arrow in the Figure 8) from the left end face of the part. The inner diameter of the characteristic section is detected by detecting the trigger state changes of D1 and D2. The inner contour step can be summarized into two states: state1 and state2. After

D1 and D2 move to the right end face, the internal spline features are determined according to the detection process of the number of teeth (Figure 6a).

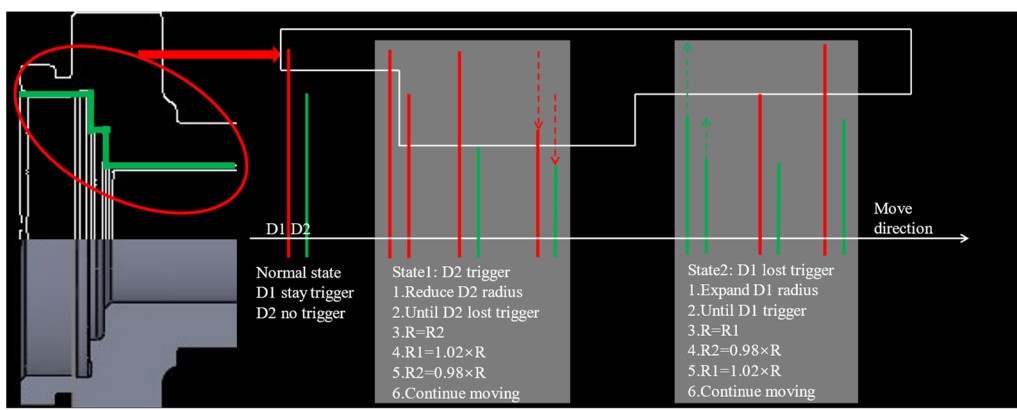

**Figure 8.** Process and mesh solid model for detecting the inner contour.

2.1.3. Mesh Solid Modelling

The key parts of agricultural machinery follow strict parametric design. This study starts with the design parameter equation of each part to fit the part contour, and the generated mesh solid structure corresponds to the real structure. For example, the involute equation is used to fit the gear. Compared with the traditional complex surface fitting algorithm, its mesh solid accuracy and calculation efficiency are higher. The mesh solid model built by the system is the core part of the virtual prototype, which solves the problem of the weak ability of the traditional virtual prototype in physical simulation and continuous collision and interference detection of multiple parts.

The arc feature can be decomposed into a mesh solid model composed of a finite number of trapezoidal elements. The gear features can be decomposed into finite tooth section elements along the axial direction to fit the gear features of helical, spiral, and conical teeth. The bright green wireframe in the following figures represents the bounding box of the part. Figure 9a is the 3D model; Figure 9b shows the built-in mesh trigger without physical simulation and collision trigger detection function; Figure 9c is the mesh solid model constructed in this paper, including the gear feature and arc feature of the inner hole.

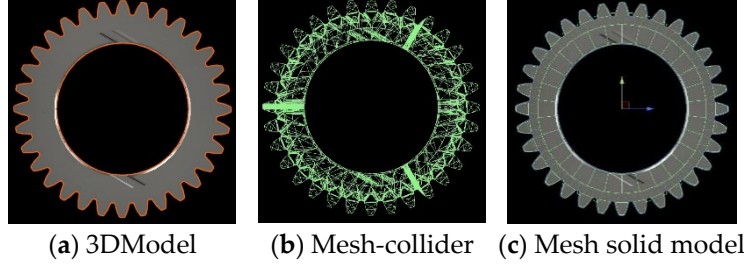

(**a**) 3DModel      (**b**) Mesh-collider   (**c**) Mesh solid model

**Figure 9.** Relationship diagram between the 3D model and the mesh solid model.

Gear Feature Mesh Solid Modelling

$$R_b = M \times Z \times cos20° /2 \tag{12}$$

$$\frac{\varphi_a - \varphi_b}{A_T} \times i + \varphi_b = \varphi_i \tag{13}$$

$$\frac{arcsin(M \times \pi/4)}{\sqrt{z^2 + y^2}} + arctany/z = \delta_r \tag{14}$$

$$t = B/A_r \tag{15}$$

$$2 \times \pi/Z = \chi \tag{16}$$

First, the algorithm divides the rolling angle range $[\varphi_b, \varphi_a]$ into finite values $\varphi_i$ according to the accuracy $A_T$ and obtains the corresponding points $\overrightarrow{P_i}$ on the involute; among them, $\varphi_a$ is the rolling angle corresponding to the intersection P2 of the addendum circle and the involute, and $\varphi_b$ is the rolling angle corresponding to the intersection P1 of the base circle and the involute. Then, the involute rotates around the origin $\delta_r$ radians, mirrors clone along the z-axis to obtain all differential contour points of the gear section and translates and clones the gear section points on the back to construct the mesh element (see gear section and mesh element). In Formula (12), $M$ is the module of the spur gear and helical gear and the module of the large end of the spiral bevel gear; $Z$ is the number of teeth of the gear; and $y, z$ are the Y and Z coordinates of the intersection of the dividing circle and the involute. $B, A_r$ are the width of the gear and the division accuracy of the tooth line, respectively. When the gear is a cylindrical spur gear, each mesh element is arranged along the axial direction (see spur gear in Figure 10). When the gear is a cylindrical helical gear or a bevel gear, each mesh element with a thickness of $t$ is arranged along the helix or spiral direction (see the black dotted line in helix gear and spiral bevel gear) to form the mesh solid model of a single gear tooth. Finally, the mesh solid model of a single gear tooth is cloned Z times, and each clone rotates $\chi$ radians around the rotation axis of the part to construct the mesh solid model of gear features (see helix gear and spiral bevel gear in Figure 11).

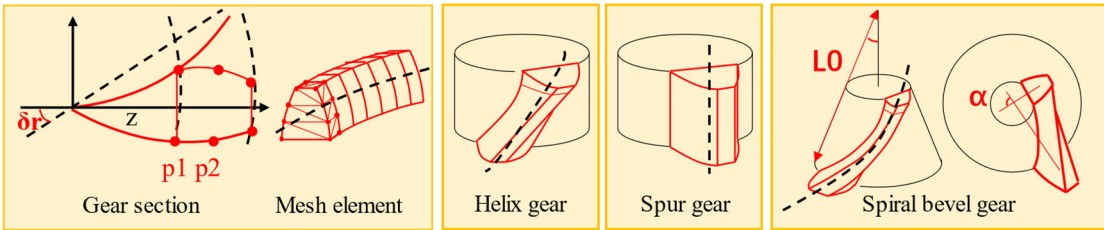

**Figure 10.** Process of mesh solid modelling of gear.

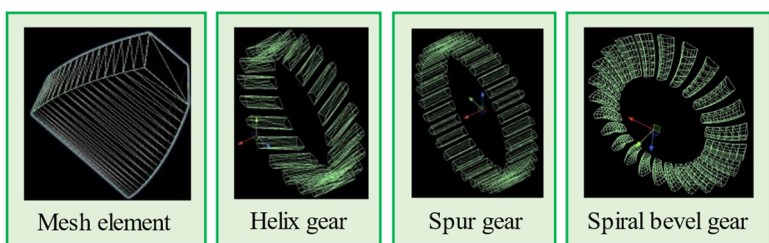

**Figure 11.** Mesh element and mesh solid model.

Arc Feature Mesh Solid Modelling

The algorithm divides the arc into broken segments and uses a finite number of trapezoidal mesh elements to fit the arc features. The fitting diagram and effect diagram of the mesh solid model are as Figure 12. The red trapezoid represents a mesh unit with an arc feature.

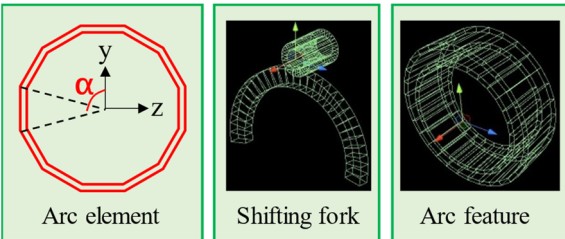

**Figure 12.** Arc feature mesh solid model.

$$\alpha_k = k \times \frac{\theta}{A_a} \tag{17}$$

$$\vec{p_i} = \left[ x_c \pm \frac{T_a}{2} \quad R \times \cos \alpha_k \quad R \times \sin \alpha_k \right]^T \tag{18}$$

The algorithm divides the arc in which the center angle is $\theta$ into $A_a$ arc elements. $\vec{p_i}$ is the vertex on the arc element, and $\alpha_k$ is the included angle between its radius line and $y$-axis. The value range of $k$ in Formula (17) is $[0, A_a]$. When $\vec{p_i}$ is on the outer contour, $R$ is the outer diameter of the contour. In contrast, $R$ is the inner diameter. $x_c, T_a$ represents the centroid X coordinate and thickness of the arc feature.

### 2.1.4. System Error Analysis
Error Analysis of the Feature Parameter Detection Process

In this study, GoogLeNet is used to classify the parts corresponding to the image according to the sample image of the part; YoloV4Tiny is used to detect all part features in the sample image according to the type of features contained in the part. Their dataset is classified according to the types of parts and their characteristics summarized in Table 1. There are about 1500 samples of each type of parts. The training set, test set, and verification set are divided according to the quantity ratio of 8:1:1 to construct the data set of GoogLeNet image classification network. This data set, according to the relationship between parts and their feature types summarized in Table 1, frames the data features in various parts' image samples, and constructs the data set of YoloV4Tiny. The schematic diagram of the data set is shown in Figure 13. Their recognition and detection accuracies are both higher than 99.6%. In this study, approximately 10,000 three-dimensional models are used to construct its training set. Some of its data and the PR curve of network training are as shown in Table 3 and Figures 13 and 14.

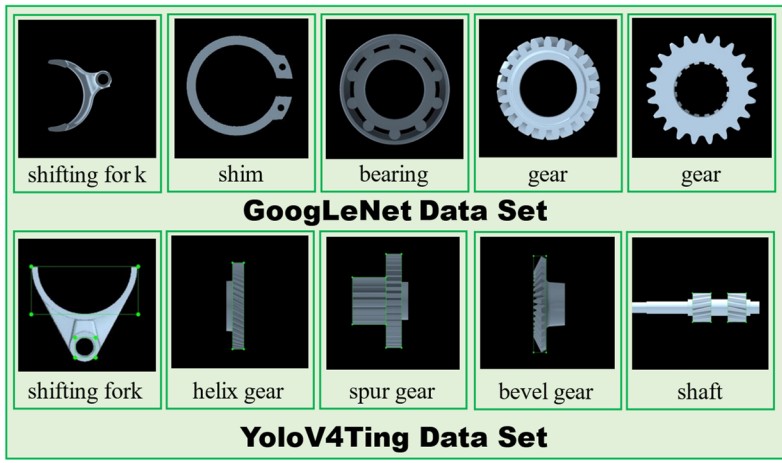

**Figure 13.** The training dataset of the deep-learning network.

**Table 3.** Deep learning network training data table.

| Deep Learning Net | Precious | Recall |
|---|---|---|
| GoogLeNet | 99.61% | 99.34% |
| YoloV4Tiny | 99.23% | 99.87% |

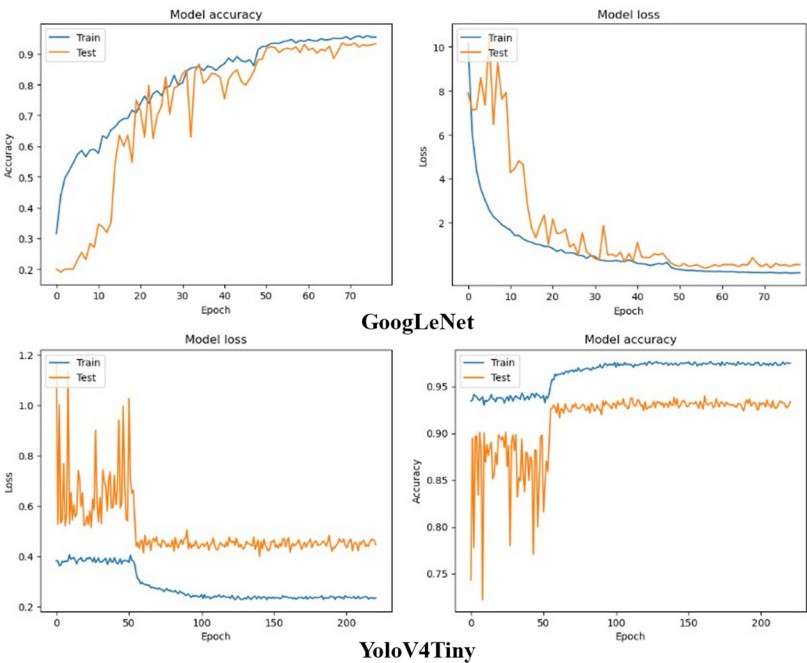

**Figure 14.** The loss and accuracy curve of the deep-learning network.

On the other hand, the error unit of the image-processing algorithm is the unit length of pixels. We randomly selected several three-dimensional models with known part parameters, compared them with the system detection data, and solved the relative error of the data. The results are as shown in Table 4.

**Table 4.** Statistical table of error verification data.

| Part Name | Part Feature Parameters | Theoretical Value | Detection Value | Relative Error |
|---|---|---|---|---|
| Second driven gear | Radius of addendum circle | 49 | 48.9997 | $6.12 \times 10^{-6}$ |
| | Tooth width | 26 | 26.0003 | $1.15 \times 10^{-5}$ |
| | Centroid of feature segment | $(-68.6, 0)$ | $(-68.5998, 0)$ | $2.92 \times 10^{-6}$ |
| | Inner arc radius | 29 | 29.0008 | $2.76 \times 10^{-5}$ |
| | Inner arc thickness | 39 | 38.9998 | $5.13 \times 10^{-6}$ |
| | Centroid of inner contour feature segment | $(-68.6, 0)$ | $(-68.5999, 0)$ | $1.46 \times 10^{-6}$ |
| Inner bore bearing of fourth driven gear | External diameter | 37 | 37.0004 | $1.08 \times 10^{-5}$ |
| | Internal diameter | 25 | 24.9999 | $4.00 \times 10^{-6}$ |
| | Width | 30 | 30.0005 | $1.67 \times 10^{-5}$ |
| | Centroid of feature segment | $(6,0)$ | $(6.0001,0)$ | $1.67 \times 10^{-5}$ |

From the data in Table 4, it can be concluded that the relative errors of some feature parameters are below $2 \times 10^{-5}$, which is better than the measurement results of the feature recognition network (see the detection results in Figure 3).

Error Analysis of Mesh Solid Modelling

The error of differential calculation will occur in the arc fitting of the mesh solid model. Increasing the values of $A_r, A_T, A_a$, which are in Formulas (13), (15), and (17), can improve its contour accuracy. Using the error analysis method of curve fitting [29–31], we selected any vector diameter in the cylindrical coordinate system to intersect the mesh solid model and the 3D model, obtained two surface contour points $(r, \theta, h), (r_r, \theta_r, h_r)$, and then solved the relative error $\sigma_i$ of their spatial position to evaluate the fitting accuracy of the mesh solid model. The average frame rate data are used to evaluate the consumption of computing resources of mesh solid models with different division accuracies. The statistical data are shown in Table 5.

$$\sigma_i = \frac{|(r, \theta, h) - (r_r, \theta_r, h_r)|}{|(r_r, \theta_r, h_r)|} \tag{19}$$

**Table 5.** Fitting error data of the part mesh solid model.

| Assembly Part Name | Assembly Type | Mesh Generation Accuracy of Assembly Parts | Average Frame Rate (Hz) | Mean Value of Relative Error of Mesh Solid Modelling |
|---|---|---|---|---|
| Third gear driving gear and input shaft | Fit of internal spline | 5 | 58.43 | 0.1368% |
| | | 10 | 55.38 | 0.1296% |
| Second driven gear and bearing | Shaft-bore fit | 5 | 57.96 | 0.0821% |
| | | 10 | 55.10 | 0.0857% |
| 2nd driven gear bearing and output shaft | Shaft-bore fit | 15 | 51.91 | 0.0596% |
| | | 20 | 48.72 | 0.0631% |

Meshing accuracy refers to the number of differential line segments fitting the curve of unit length. When its value is larger, the fitting accuracy is higher. On the premise of ensuring that the fitting error is lower than ±0.1 mm of the machining standard, when the division accuracy is taken as 5, the fitting error is about 0.1%, equivalent to ±0.001 mm. At this time, the operating frame rate of the system is about 58 Hz; when the division accuracy is taken as 10, the reduction of fitting error is not significant, and the operating frame rate drops to 55 Hz; when the division accuracy is greater than 10, the system runs stuck.

The conclusion shows that the best division accuracy of the mesh solid model is 5. On the premise of ensuring the accuracy of the design process, the mesh solid model avoids the complex surface fitting process of traditional algorithms and constructs a virtual prototype similar to the actual structure.

### 2.2. Construction of the Assembly Datum Layer

The traditional virtual assembly actually uses geometric logic to describe the general assembly behaviour, which requires the operator to have high geometric language analysis and judgement ability, lacking intuition and authenticity. This paper simplifies the regular assembly behaviour of key parts of agricultural machinery. The operator uses VR for virtual assembly to improve the interactivity and authenticity. The system uses simple assembly behaviour, consistent with the actual operation, to assemble, ensuring the consistency of users' operation and reducing the difficulty of virtual assembly.

The system uses the geometric logic of the assembly datum to define the assembly behaviour of parts. The assembly datum axis and assembly reference plane are located according to the part rotation axis and step surface of the mesh solid model, which are used for shaft hole alignment assembly and part assembly positioning, respectively, and their positioning error is the same as that of the mesh solid model. Designers screen the incorrect assembly benchmark through the visual interface to improve the fault tolerance rate of the system (see Figure 15).

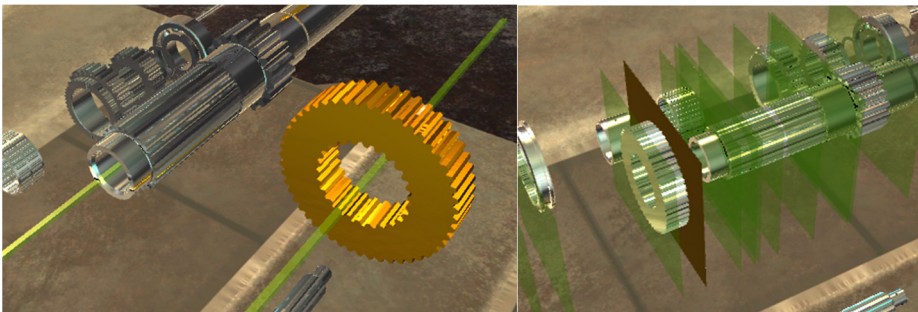

**Figure 15.** Assembly datum axis and face.

*2.3. Construction of the Data Layer and 3D Model Layer*

The data layer is a node where each part of the data is calculated and stored independently. Multiple hash tables are set in the script of this node to store the data from the mesh solid model and knowledge base. During the assembly process, the system continuously calls the node data for knowledge reasoning and feeds back the assembly test data in real time. The sub-nodes of the 3D model layer of the virtual prototype correspond to the 3D model of the part, which improves the data processing efficiency of simultaneous feedback from multiple parts.

## 3. Construction and Application of Knowledge Base

The design parameters and calculation logic of the traditional intelligent design system have been encapsulated in the program, so it is impossible to change the design standards and calculation logic according to the changeable requirements of small batch production. The knowledge base built by this system is completely inputted and defined by users, and different calculation logic and design standards can be specified according to different production needs.

The system constructs the mesh solid model of a virtual prototype with various design and test functions. It can call the design process data and script logic commands in the knowledge base to control multiple parts for multiple design tests at the same time. The virtual prototype of each part can detect and feedback its own test data independently. At the same time, the knowledge base data are dynamically generated by data tables.

Compared with the traditional agricultural machinery intelligent design system, which completes various design and test functions and highly encapsulates design process data in different commercial plug-in interfaces, this system has an independent, automatic, and synchronous single part data processing function and dynamically updated user-defined design process data, which is more in line with actual production needs.

The knowledge base is divided into five parts. Among them, the knowledge base of assembly, part classification, and assembly sequence evaluation calculates the data according to the judgement logic and calculation process in the script. Tolerance fit, interference analysis, and interchangeability knowledge base need to use feature radius and accuracy grade to retrieve the table data under the root directory of the system, and then use script for operation.

*3.1. Knowledge Base for Virtual Assembly*

Virtual assembly controls the geometric calculation of multi-object interactions by means of dynamics or kinematics. The system carries out human–computer interactions using VR equipment, calls the mesh solid model layer for collision feedback [32,33], simplifies the assembly logic into two assembly behaviours: datum axis alignment and datum plane

alignment [34], and uses the kinematic equation to describe the motion of the object. Finally, the system sets the range threshold and adjusts the part to the assembly pose [21,35].

$$\theta = arctan\left(\frac{\vec{a_b} \cdot \vec{a_p}}{\left|\vec{a_b}\right| \cdot \left|\vec{a_p}\right|}\right) \tag{20}$$

$$\vec{L_p} = \vec{f_p} - \vec{f_b} \tag{21}$$

$$\vec{P}' = R_p \cdot \vec{P} + L_p \tag{22}$$

In the formula, $\theta$ is the included angle of the assembly datum axis between the assembly part and assembly body; $\vec{a_p}, \vec{a_b}$ and $\vec{f_p}, \vec{f_b}$ are the direction vectors of the assembly datum axis and the normal vector of the assembly datum plane of the parts and assembly body, respectively; $\vec{L_p}$ is the displacement vector of the part; $R_p$ is the rotation matrix rotating around a specific axis with the centroid of the part as the origin; and $\vec{P}, \vec{P}'$ are the position vectors before and after part assembly.

*3.2. Knowledge Base for CAD*

In the process of CAD, the tolerance zone, fitting form, and assembly sequence of parts are designed according to the mechanical manufacturing process standard. Then, combined with the production needs of the enterprise, the design evaluation standard evaluates the design scheme and guides the computer-aided design. Among them, the data table of the knowledge base is the .csv binary table file in the root directory of the system. The system retrieves the data of the tolerance table, dimension deviation table, sequence evaluation data table, and interchangeability data table with the parameters of feature diameter, accuracy grade, and evaluation index, inputs the corresponding script for calculation, outputs it to the system, and updates the data of the virtual prototype. It should be emphasized that every time the system starts, it traverses the .csv file and dynamically generates a hash table, and the data in the .csv file can be updated. This solves the problem that traditional intelligent design platforms face, in that it cannot change the calculation logic and production data encapsulated in the system to meet the changing design requirements of small batch production. The knowledge base structure is shown in Figure 16.

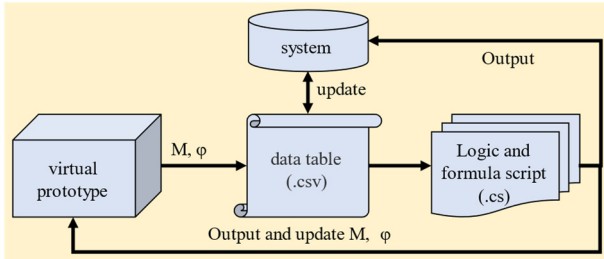

**Figure 16.** Data processing of the knowledge base.

3.2.1. Knowledge Base of Part Basic Classification

This part of the knowledge base data is encapsulated in the script code. First, the system converts the 3D model into triangular patch structure according to the process of Section 2.1.1. FBX model format (see Figure 17 format conversion). Then, the system reconstructs the centroid coordinate system of the model and adjusts the zoom ratio (see Figure 17, reconstructs the centroid coordinate system) and samples according to the fixed pose of the centroid of the model to obtain the sample image (see Figure 17, sampling). Among them, the system classifies axis parts and general parts according to the length

width ratio of the model bounding box after reconstructing the centroid coordinate system (see Figure 17 part classification). Finally, the system inputs the sample image of general parts to GoogLeNet to classify general parts such as gears and bearings; then inputs all classified part sample data into OpenCV image-processing algorithm to detect all feature parameters, and inputs them into YoloV4tiny to detect the feature types contained in each category of parts (see Figure 17 feature parameter extraction).

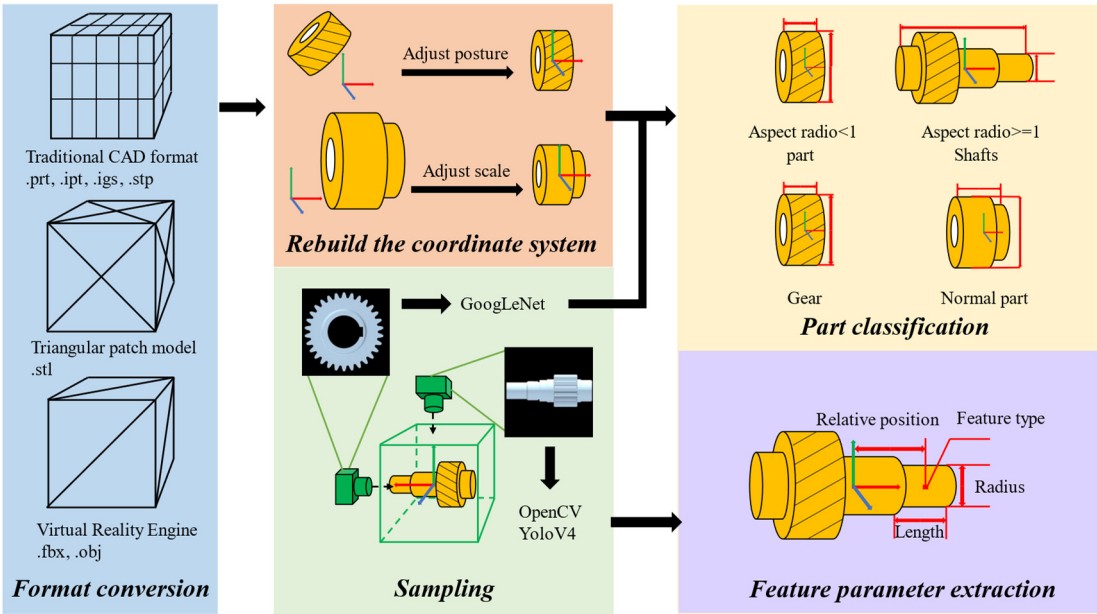

**Figure 17.** Technology roadmap.

### 3.2.2. Knowledge Base of Assembly Sequence and Evaluation

The assembly main body of key parts of agricultural machinery is generally the shaft, and its assembly parts are stepped or spline shaft segments, corresponding to each feature on the mesh solid mode [36]. The system matches the feature with the feature parameters of the part to generate a variety of assembly sequences. This part of the knowledge framework is shown in Table 6.

**Table 6.** Part feature matching knowledge framework.

| Assembly Body Feature | Combination of Multilevel Assembly Parts | Matching Basis |
|---|---|---|
| Spline segment | Internal spline gear-Meshing sleeve<br>Internal spline gear<br>Snap ring | Modulus |
| Arc segment | Bearing/Shaft sleeve<br>Bearing/Shaft sleeve-Gear<br>Snap ring | Diameter |
| Fork slot | Mesh sleeve-Shifting fork | Diameter |

Taking the assembly main body axis in the upper part in Figure 18 as an example, the algorithm takes the *x*-axis (red arrow) of the assembly main body as the positive direction and uses the assembly feature parameters with frames A and B to match the parts. For example, the assembly serial numbers of Parts C1, D1, D2, E1, and E2 are (+2.1), (+5.1), (+5.2), (+7.1), and (+7.2), respectively, where D2 (+5.2) represents the part of the second layer of the fifth part combination of forward assembly (Box D represents a part combination).

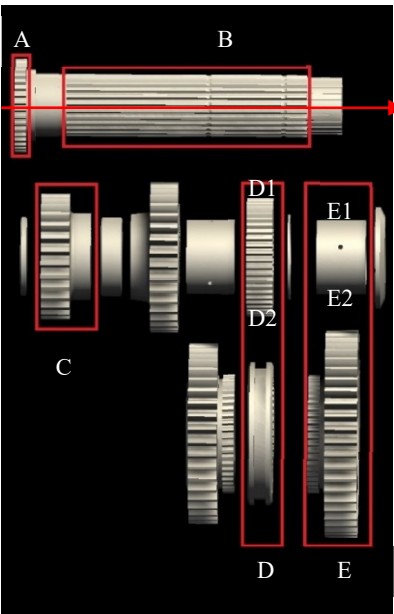

**Figure 18.** Assembly sequence diagram.

We count the actual production data of the workshop, such as the maximum and minimum values of the assembly time, the moving distance of the assembly personnel in the workshop, and the replacement frequency of assembly tools and other indicators. Then, we divide the score range of 100 with a step of 10 points and store it in the knowledge base. The system calls the assembly index values of various parts within this framework to calculate the evaluation value of each assembly sequence [23]. Part of the data of the knowledge base is shown in Table 7.

**Table 7.** Assembly sequence evaluation knowledge framework.

| Part Type | Installation Tools | Assembly Time | Assembly Difficulty |
| :---: | :---: | :---: | :---: |
| Driven gear | grease | d3 | e3 |
| Driving gear | grease | d3 | e3 |
| Meshing sleeve | grease, screwdriver | d4 | e4 |
| Shaft assembly subassembly | lifting arm | d5 | e5 |

$$i \times \frac{100}{k_j} = \alpha_i^j \tag{23}$$

$$T_k = \sum \alpha^j \tag{24}$$

Formula (23) solves the assembly time and assembly difficulty evaluation value $\alpha_i^j$ of the ith level of the jth evaluation index of the assembly sequence. $k_j$ is the number of index division levels. Formula (24) is the total evaluation value of the kth assembly sequence.

3.2.3. Knowledge Base of the Tolerance Fit and Interference Analysis

The interference amount to be tested in the tolerance fit design is mainly at the shaft hole fit, including the interference fit amount and interference amount caused by improper design. The user sets the tolerance grade of the part and retrieves the tolerance zone data according to the feature diameter. Then, the system solves the size deviation and limit size of the matched part features according to the set matching benchmark system. Next, the system updates the parameters of the solid model layer of the part mesh to result in dimensional deviation. Finally, when the user carries out virtual assembly, the system

retrieves the data within the knowledge framework according to the diameter of the feature and the set accuracy level to analyze whether the interference of the part is reasonable in real time. Part of the knowledge framework is shown in Table 8.

**Table 8.** Tolerance zone.

| Class<br>Diameter (mm) | IT4<br>(μm) | IT5<br>(μm) | IT6<br>(μm) |
|---|---|---|---|
| φ18–φ30 | 6 | 9 | 13 |
| φ30–φ50 | 7 | 11 | 16 |
| φ50–φ80 | 8 | 13 | 19 |

The system transforms the color of the mesh cell body in the part mesh solid model, marks the interference position, extracts its centroid, and produces the scatter diagram of interference distribution, which intuitively reflects the distribution of interference quantity, as shown in Figure 19.

$$\eta_i = min\left\{ R - r - \sqrt{\left(y_{ij} - y_{ic}\right)^2 + \left(z_{ij} - z_{ic}\right)^2} \right\} \tag{25}$$

$$\eta_{yz} = max\{|\eta_i|\} \tag{26}$$

$$\eta_x = \left| |x_{c1} - x_{c2}| - \left(\frac{T_1 + T_2}{2}\right) \right| \tag{27}$$

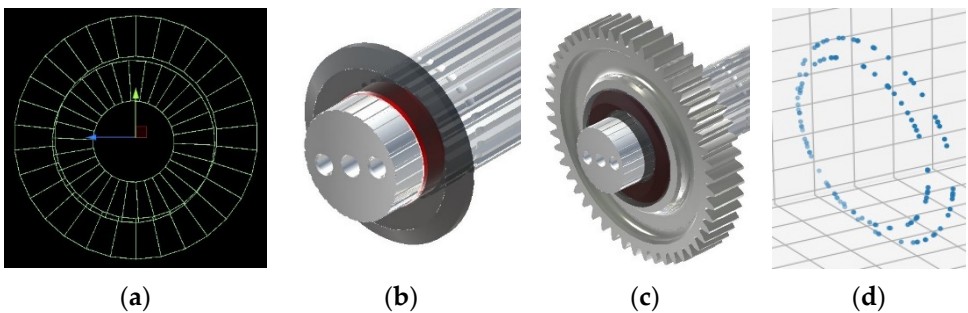

| (a) | (b) | (c) | (d) |
|---|---|---|---|

**Figure 19.** Interference data. (**a**) Schematic diagram of grid interference. (**b**) Shaft hole interference. (**c**) Surface interference. (**d**) Interference point cloud data.

$\left(x_{ij}, y_{ij}, z_{ij}\right)$ is the centroid coordinates of the jth mesh cell on the assembly part feature that interferes with the ith mesh cell on the feature of the assembly body. $\left(x_{ic}, y_{ic}, z_{ic}\right)$ is the centroid of the ith mesh cell body in which interference occurs on the feature of the assembly body. $\eta_i$ is the interference value of the ith mesh cell on the feature of the assembly body; when it is negative, the smaller the value is, the greater the interference; when this value is positive, there is no interference. Take the maximum value of all mesh elements that interfere with the assembly body as the radial interference $\eta_{yz}$. The axial interference $\eta_x$ is calculated by the distance between the centroid X coordinate $x_{c1}, x_{c2}$ of the interference feature and the feature thickness $T_1, T_2$.

### 3.2.4. Knowledge Base of the Parts Interchangeability

The production process statistics of multiple workshops within this knowledge framework is used to calculate the matching interference law of different batches of parts and generate the interchangeability matching scheme of products in each workshop [7,15], which is used to estimate the production cost and modify the tolerance matching scheme. For application examples, see Section 4.2.

### 3.2.5. Fault Tolerant Processing of the System

In the process of knowledge reasoning and building of a solid mesh model, the following errors will occur with low probability: (1) feature recognition error; (2) assembly sequence generation error. The system sets the mesh solid model visualization interface and assembly sequence visualization interface, on which the setter can view the speculation results and manually change the data. Users can reconstruct the mesh solid model by modifying the script parameters mounted on the mesh solid model layer, as well as modify the assembly sequence by modifying the assembly sequence number in the data layer of each part.

## 4. System Application Example

Using the bearing selection scheme data of transmission components of an agricultural machinery gearbox, designed and produced by an enterprise, the recommended bearing model is generated after automatically completing the construction of the virtual prototype, and the accuracy of the system is verified by comparison with the actual process data. Finally, the system performance is evaluated by the software evaluation of the third-party organization. The field data measurement is shown in Figure 20.

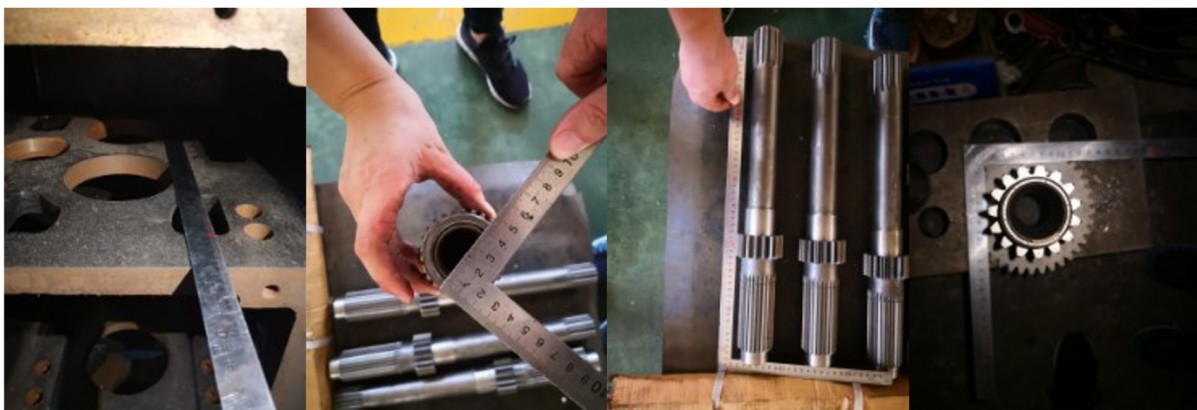

**Figure 20.** Measurement process of some parts.

### 4.1. Construction of Virtual Prototype and Generation of Assembly Sequence

The assembly sequence visualization interface and the assembly sequence number of each part are shown in Figure 21, as well as the assembly sequence evaluation data in Table 9.

**Table 9.** System generated assembly sequence scheme.

| Assembly Sequence Plan | Score | Optimal Scheme |
|---|---|---|
| Plan A: +1.1, +2.1, −1.1, −2.1, −3.1, −3.2, −4.1, −4.2 | 52 | |
| Plan B: −1.1, −2.1, −3.1, −3.2, −4.1, −4.2, +1.1, +2.1 | 46 | √ |

The reason why Plan B is chosen is that the last part of the right half and the first part of the left half needs to be installed with a tail hook wrench. If the right half is installed first and then the left half is installed, the time of repeated tool access will be saved.

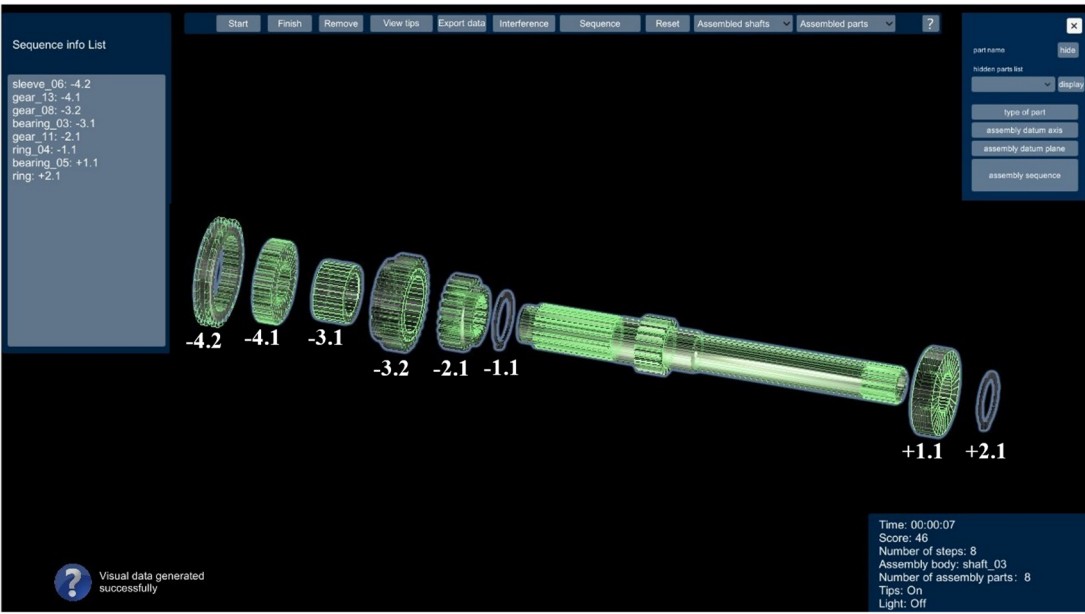

**Figure 21.** Solid model of the sub−assembly part mesh.

*4.2. Generation of Assembly Interference and Matching Schemes and Incomplete Interchange Schemes*

We investigated and obtained the production process statistics of workshops I and II (Table 10), updated the mesh solid model layer of the virtual prototype for the updated data statistics (Table 11), and counted the interference data and interchangeability matching data after virtual assembly (Table 12).

**Table 10.** Production process statistics table in the knowledge base.

| Dimensional Tolerance (mm) / Accuracy Class | Workshop I | | Workshop II | |
|---|---|---|---|---|
| | IT6 | IT7 | IT6 | IT7 |
| φ30–φ50 | 14 | 23 | 16 | 25 |
| φ50–φ80 | 17 | 28 | 19 | 30 |
| φ80–φ120 | 20 | 33 | 22 | 35 |
| Maximum allowable fit clearance | | 10 | | |
| Maximum allowable fit interference | | 10 | | |

**Table 11.** Fit tolerance data of key parts.

| Part Features Name | Tolerance Zone | Basis Fit Systems | Limit Size (mm) |
|---|---|---|---|
| φ110 Case input shaft locating bearing hole | J7 | shaft-basis system of fits | D min = 109.9825 D max = 110.0175 |
| φ50 Input axle box body positioning bearing journal | h6 | hole-basis system of fits | d min = 49.984 d max = 50.000 |
| φ110 Locating bearing hole of secondary transmission axle box | J7 | shaft-basis system of fits | D min = 109.9825 D max = 110.0175 |
| φ70 Positioning bearing journal of secondary transmission axle box | h6 | hole-basis system of fits | d min = 69.981 d max = 70.000 |

**Table 12.** Product scheme data.

| Part Number | Average Size of the Workshop 1 | Workshop 1 Average Interference/ Clearance (mm) | Average Size of the Workshop 2 | Workshop 2 Average Interference/ Clearance (mm) | Recommended Bearing Parameters |
|---|---|---|---|---|---|
| φ110H1 φ50S1 | φ109.9877 φ49.9935 | −0.0055 0.0021 | φ109.9867 φ49.9936 | −0.0087 0.0025 | 31310P0 |
| φ110H2 φ70S1 | φ109.9970 φ69.9881 | −0.0053 0.0024 | φ109.9864 φ69.9924 | −0.0090 0.0031 | 32014P0 |

According to the data in Table 12, the product size of workshop 1 is small and that of workshop 2 is large. The system suggests that the bearing with a larger mean limit size should be matched with the product of workshop 1, while the bearing with a smaller mean value should be matched with the product of workshop 2. Then, according to the given data, we select three types of bearings in the table for virtual assembly. Finally, the interference data of the system statistics show that it meets the proposed fit clearance requirements.

### 4.3. System Performance Evaluation

The cooperative enterprise of the project funded by this paper employs a third-party evaluation company to conduct software evaluation. Using a computer configured with 32 GB memory, a CPU of interi59400f, and graphics card of NVIDIA 2070 Super as the test platform, the time and assembly operation error rate of four professional evaluation personnel (No. 1–4) completing the intelligent design task of this chapter, using the intelligent design system of SolidWorks and this system are counted as the performance indicators of this system. Each professional completes the design step and assembly step five times (see Figure 22). We count the average operation time, assembly error rate, and computer resource consumption rate of each person; the results are shown in Figure 23.

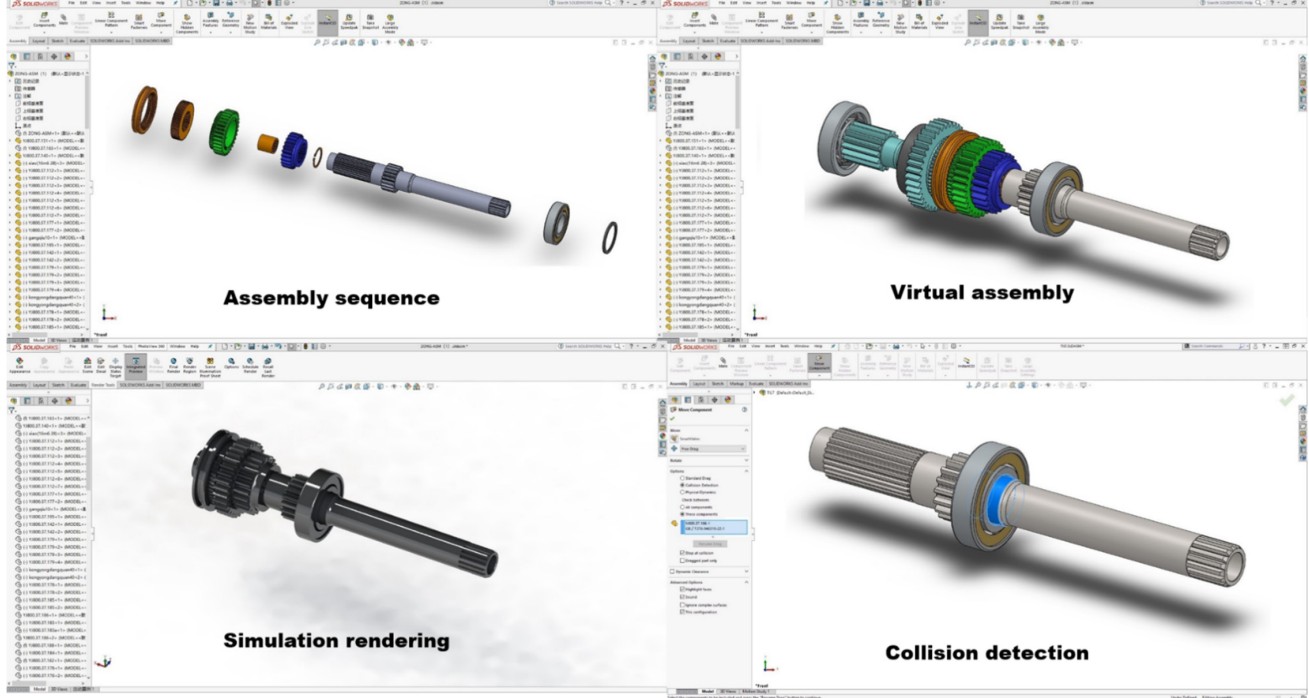

**Figure 22.** Screenshot of the operation process.

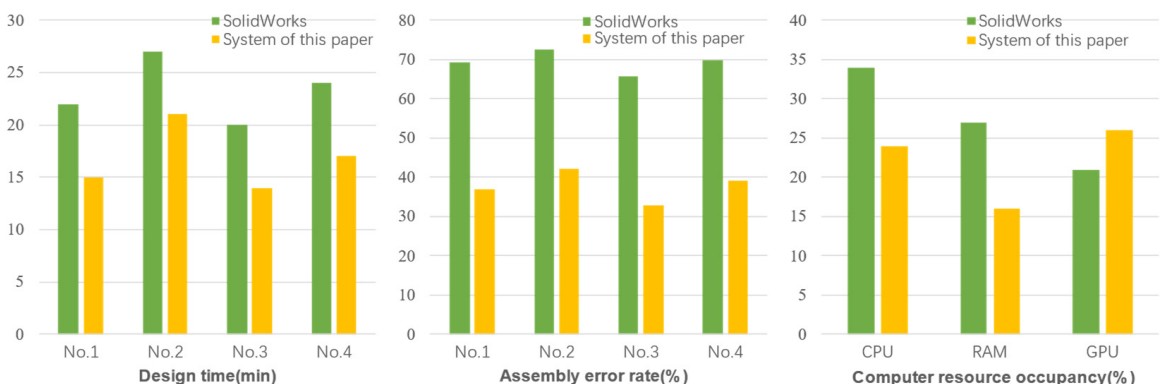

**Figure 23.** System performance comparison data.

From the evaluation data, the design efficiency of this system is improved by approximately 20% compared with that on the SolidWorks platform, the operation error rate is reduced by approximately 40%, and the resource consumption of CPU and ram is reduced by approximately 30%. Moreover, the cost of using the commercial version of SolidWorks and its plug-in function package is approximately 200 thousand yuan, which is much higher than the selling price of the commercial development platform Unity3DPro used in this system.

*4.4. Conclusions*

The system can generate the analysis data required for production design and can complete the data analysis of the complete assembly scheme and interchangeability scheme, automatically and consistently. On the premise of ensuring that the design data meet the design accuracy, the system generates a preliminary design scheme reference meeting the production process requirements. In the early stages of part design, the system can use limited data to generate a more comprehensive design scheme, improve the production design efficiency of small batches and multiple changes, reduce the error rate of manual operation, and reduce the cost of purchasing intelligent design software and computer hardware configuration.

**5. Summary**

This research combines deep learning and image-processing technology to build a virtual prototype with complete functions and a novel structure. Combined with knowledge reasoning and data-driven technology, a special knowledge base is constructed, which is integrated into a flexible, fully functional, and practical agricultural machinery intelligent design system in a virtual reality engine.

(1) Based on the existing 3D model, this study automatically batch constructs a general virtual prototype of key parts of agricultural machinery, integrating the functions of interference detection, cooperation analysis, and physical simulation. In the early stages of design for small and medium-sized enterprises, the limited existing design data can be reused, and new parameter schemes can be obtained according to the new production process, which can significantly reduce the production cost during the early stages of reuse design. The data volume of the lifting system can also be extended to the fields of automobiles, aircraft parts, and so on.

(2) The system knowledge base takes the dynamically updated tabular data as the knowledge framework for knowledge reasoning; additionally, its flexibility is much higher than the product testing function encapsulated in traditional CAD software. Through the combination of various functions of knowledge reasoning, the intelligent design process of various parts is completed. Compared with the traditional CAD software, it obtains various analysis data by calling various plug-in interfaces and then integrates them into a new design scheme, which has higher efficiency and a more coherent

design process. Each plug-in of traditional CAD software needs to be purchased separately, while this system is based on a virtual reality engine to freely integrate a variety of functions with lower cost.

(3) With the help of a data visualization interface, the system provides an operation interface for designers to modify the wrong mesh solid model feature parameters and screen knowledge reasoning data to improve the fault tolerance of the system. At the same time, the feature recognition network can be used to extract more useful data from CAD drawings of reuse design.

(4) The system is oriented to the production application of small batch and changeable processes in small and medium-sized agricultural machinery equipment enterprises and overcomes the disadvantages of low development freedom, high cost of module function integration, and low module compatibility of large-scale intelligent design CAD systems.

There are still deficiencies in the accuracy of dynamic simulation data and finite element analysis function of this system; the finite element function and precision matrix operation program library will be further integrated in the future. Additionally, we will learn from the functions of traditional CAD intelligent design systems to further improve the practicality of the system.

**Author Contributions:** Conceptualization and methodology, X.Z. and C.L.; validation, C.L., P.Z. and J.L.; data curation, C.L.; writing—original draft preparation, C.L., Y.T., P.Z., G.L. and J.L.; writing—review and editing, X.Z., G.L. and Y.P.; visualization, C.L. and G.L.; supervision, X.Z.; funding acquisition, Y.P., G.L. and X.Z. All authors have read and agreed to the published version of the manuscript.

**Funding:** This study was funded by The National Key Research and Development Program of China (2017YFD0700100) and Provincial Science and Technology Department Funding Project of Jiangxi, China (Grant No.20212ABC03A27).

**Institutional Review Board Statement:** Not applicable.

**Informed Consent Statement:** Not applicable.

**Data Availability Statement:** Not applicable.

**Acknowledgments:** We sincerely acknowledge the project funding support. We are also grateful for the efforts of all our colleagues.

**Conflicts of Interest:** The authors declare that they have no conflict of interest.

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
