# Peer review of "A Novel Agricultural Machinery Intelligent Design System Based on Integrating Image Processing and Knowledge Reasoning"

_applsci, doi:10.3390/app12157900_

Round 1

Reviewer 1 Report

General comments:

1.Several grammatical mistakes, convoluted sentences and lack of clarity in explanation

2.In the abstract and introduction a very general solution is suggested whereas the authors seemed to have focused heavily on identifying gear parameters, reusing theory heavily from machine dynamics and kinematics, the aspect of intelligent design is highly suppressed.

3.line 24-27 in the abstract- where is the justification later in the paper? there is no rigorous benchmark/dataset/performance comparison across different techniques

4. how can I believe that you solution - "improves the efficiency of intelligent design in key parts of 25 agricultural machinery by approximately 20%, reduces the operation error rate of personnel by approximately 40% "?

5.Are you going to make the dataset public?

6.Is open-source code available for your solution?

Example grammar mistakes:

1."Agricultural machinery intelligent is the inevitable direction" - intelligence

2.line 46 starting sentence- start with "In this work we analyze ..."

Specific comments:

1.section 2.1.1 lacks clarity in explanation- what is batch format? what is secondary development interface?

2.the data normalization aspect needs to be explained in more detail with a simple example/ how are you scaling the part to the appropriate size?

3.line 171,172 - "and avoids the accurate data measurement function of deep learning instability"- meaning not clear what is deep learning instability and why will it be caused?

4.line 185-186 not clear what you mean

5.line 227-228 - "The feature recognition network has poor accuracy in identifying a large number of features."- what network are we talking about here? so far you have only mentioned about canny detector which is not a neural network

6.section 2.1.4- why have you specifically chosen GoogLeNet and YoloV4Tiny? The formulation of test dataset and validation dataset are not clear. 

7.The data annotation process is not clear. Metrics used are insufficient and general formulation of the problem is still unclear- are you implementing key-points detection? object detection? or plain image classification? What is the type of the learning problem involved?

8.suggest moving figure 17 near introduction and providing a solid case specific step by step description along with the figure

Reviewer 2 Report

The manuscript presents a path of knowledge of a problem and of the construction of a solution through different information sources. The different steps for the construction of a virtual prototype are analyzed. It is an exhibition of a state of the art of the knowledge necessary for the preparation of mechanical parts verified in a virtual way to highlight the possible tolerances and the possible problems that can occur in the real construction. The text was written in a clear way and accompanies the reader in the problems useful for the overall understanding of the study.

The research conducted may have good potential to contribute to knowledge in the sector. I have no changes to suggest to improve the text.

Author Response

Thank you very much for your review!

Reviewer 3 Report

The manuscript deals with a novel intelligent design system for agricultural machinery based on the integration of image processing and knowledge reasoning which is interesting. It is relevant and within the scope of the journal.

How did you find the optimum mesh number? Did you perform a mesh sensitivity analysis?

The article contains an excessive number of figures; some can be given in the appendix or supplementary files.

This paper does not contain a discussion section and should be provided with more details and justifications. 

I strongly recommend that the authors discuss the difference between their work and the studies previously conducted in the literature.

Summary section should be thoroughly revised. Be clear and concise. Some sentences are redundant. Summarize the main points.

In the conclusion section, the limitations of this study suggested, that improvements in this work and future directions should be highlighted.

There are some occasional grammatical problems within the text. It may need the attention of someone fluent in the English language to enhance the readability.

Author Response

请参阅附件。

Round 2

Reviewer 1 Report

The authors have corrected the draft according to the suggestions, and have provided evidence for their performance, and also cited sources where data and code is available. Therefore, I can recommend publishing.

Reviewer 3 Report

I have no more comments. This paper is well written and can be accepted in this version.